# Identification and Analysis of a Four-Gene Set for Diagnosing SFTS Virus Infection Based on Machine Learning Methods and Its Association with Immune Cell Infiltration

**DOI:** 10.3390/v15102126

**Published:** 2023-10-20

**Authors:** Tao Huang, Xueqi Wang, Yuqian Mi, Tiezhu Liu, Yang Li, Ruixue Zhang, Zhen Qian, Yanhan Wen, Boyang Li, Lina Sun, Wei Wu, Jiandong Li, Shiwen Wang, Mifang Liang

**Affiliations:** 1National Key Laboratory of Intelligent Tracking and Forecasting for Infectious Diseases (NITFID), Institute for Viral Disease Control and Prevention, China CDC, Beijing 102206, China; thomas_ht@163.com (T.H.); liutiezhu1988@163.com (T.L.); zhangrx2020@126.com (R.Z.); m17352929533@163.com (Z.Q.); ryanwen94@163.com (Y.W.); by_li@foxmail.com (B.L.); linasun@yeah.net (L.S.); wuwei@ivdc.chinacdc.cn (W.W.); ldong121@126.com (J.L.); 2Capital Institute of Pediatrics, Beijing 100020, China; iwangxueqi@gmail.com; 3Shanxi Academy of Advanced Research and Innovation, Taiyuan 030032, China; mi_yq@saari.org.cn; 4Chongqing Research Institute of Big Data, Peking University, Chongqing 400039, China; yeli7068@outlook.com

**Keywords:** SFTS, SFTS acute phase, machine learning, LASSO–Cox, immune cells infiltration

## Abstract

Severe Fever with thrombocytopenia syndrome (SFTS) is a highly fatal viral infectious disease that poses a significant threat to public health. Currently, the phase and pathogenesis of SFTS are not well understood, and there are no specific vaccines or effective treatment available. Therefore, it is crucial to identify biomarkers for diagnosing acute SFTS, which has a high mortality rate. In this study, we conducted differentially expressed genes (DEGs) analysis and WGCNA module analysis on the GSE144358 dataset, comparing the acute phase of SFTSV-infected patients with healthy individuals. Through the LASSO–Cox and random forest algorithms, a total of 2128 genes were analyzed, leading to the identification of four genes: ADIPOR1, CENPO, E2F2, and H2AC17. The GSEA analysis of these four genes demonstrated a significant correlation with immune cell function and cell cycle, aligning with the functional enrichment findings of DEGs. Furthermore, we also utilized CIBERSORT to analyze the immune cell infiltration and its correlation with characteristic genes. The results indicate that the combination of ADIPOR1, CENPO, E2F2, and H2AC17 genes has the potential as characteristic genes for diagnosing and studying the acute phase of SFTS virus (SFTSV) infection.

## 1. Introduction

Severe fever with thrombocytopenia syndrome (SFTS) is an acute natural focal disease caused by Dabie bandavirus (DBV), previously known as severe fever with thrombocytopenia syndrome virus (SFTSV). The disease is primarily found in mountainous and hilly areas, with a higher incidence during the summer and autumn seasons and is often associated with tick bites. SFTS is characterized by symptoms such as fever, decreased white blood cell and/or platelet counts, lymph node enlargement, fatigue, and gastrointestinal symptoms. Elderly patients, those with underlying health conditions, or those who delay seeking medical attention are at a greater risk of severe illness, and critically ill patients may die due to multiple organ failure [1,2,3]. Currently, six countries (China, South Korea, Japan, Vietnam, Myanmar and Thailand) in Asia have reported cases of the virus [4,5,6,7,8,9]. China has the largest number of confirmed infections among these countries, and the number of cases is increasing every year. This is because SFTSV can be transmitted not only by tick bites [3], but also through human-to-human contact [10,11,12]. In some cases, there have been reports of transmission through contact with the blood of patients who have a high viral load [13,14], and there have also been reports suggesting that some veterinarians and pet owners in Japan have been infected with SFTSV from SFTS animals [15]. SFTSV infection is characterized by acute onset, rapid progression, and high mortality, with hospitalized patients having a mortality rate as high as 12–30% [16]. Currently, the pathogenic mechanism of SFTSV infection is not fully understood, and clinical manifestations lack specificity and can vary widely. Additionally, there is no vaccine or effective clinical intervention to prevent SFTSV infection [17]. As a result, SFTSV has become a significant global public health concern [18], with the World Health Organization (WHO) listing it as a priority infectious disease in 2017 alongside Ebola and Lassa fever [19]. Therefore, studying the specific diagnosis and the differences in immune function of SFTS is of great significance for controlling the occurrence and development of the disease. 

In recent years, with the rapid development of sequencing technology and the updating of mass spectrometer instruments, sequencing depth and sensitivity have significantly improved. At the same time, the cost of sequencing has also rapidly declined. In the coming years, there may be more deep sequencing projects focusing on studying the hosts infected by viruses. Therefore, the use of scientific and effective algorithms, such as the machine algorithm mentioned in this paper, can better help us quickly and efficiently retrieve key feature genes from a large amount of data information. Thus, in addition to the existing technical means, it is equally important to use bioinformatics analysis methods to identify more effective biomarkers for studying the pathophysiological mechanisms of SFTS. Based on this, in this study, bioinformatics analysis methods were used to re-analyze the gene expression dataset from the transcriptional analysis of blood samples from acute SFTS patients with a definite clinical background in the Gene Expression Omnibus (GEO) database. Two different machine learning algorithms were used to select the characteristic genes of SFTSV acute phase patients from the differentially expressed genes obtained. Additionally, the study utilized the currently popular cell type analysis tool named CIBERSORT to investigate and analyze the infiltration of immune cells in the SFTSV acute phase based on the identified characteristic genes. The aim of this study is to provide more reference basis for the research on gene biomarkers for the diagnosis and prognosis of the SFTS acute phase patients, and to provide new research ideas and directions for future SFTS research.

## 2. Materials and Methods

### 2.1. Data Source

The original dataset of GSE144358 (platform: GPL20795 HiSeq X Ten (Homo sapiens)), obtained from the Gene Expression Omnibus (GEO) (http://www.ncbi.nlm, (Accessed on 1 June 2023)) database, includes 37 acute phase samples and 21 samples from non-infected, healthy humans. R software (Version R-4.3.1 for Windows) was used to randomly select 80% and 20% of the data as test and validation sets, respectively. This means that 30 acute samples and 17 non-infected healthy human samples were used for the test set, while 7 acute samples and 4 non-infected healthy human samples were used for the validation set. The detailed flowchart of this study is shown in (Figure 1).

### 2.2. Principal Component Analysis 

In this study, the UMAP function in the R software UMAP package (version 0.2.7.0) was used for analysis. First, the expression spectrum was z-scored, then the UMAP function was used to reduce the dimensionality and obtain the matrix result after dimension reduction.

### 2.3. DEGs Identification 

Limma (Linear Models for Microarray Data) is a differential expression screening method based on generalized linear models. In this study, we used the limma package (version 3.40.6) in R software for differential analysis [20]. The differentially expressed genes (DEGs) between the acute phase sample cohort and the non-infected healthy group were analyzed. The resulting expression profile dataset was first subjected to multiple linear regression using the lmFit function. Then, the eBays function was used to calculate the reduction of statistics, applying log-odds differential expression of the empirical Bayes moderation to the common value standard error, and finally determining the differential significance of each gene. In this analysis, we used FDR 2.0 to generate a DEG volcano map, displaying the top 100 genes through a heat map (top 50 up-regulated and top 50 down-regulated).

### 2.4. Function and Pathway Enrichment Analysis 

For GO enrichment analysis, we used org.Hs.eg.db (version 3.1.0) in R software for gene GO annotation as the background. KEGG functional enrichment analysis was performed using the KEGG rest API (https://www.kegg.jp/kegg/rest/keggapi.html, (Accessed on 1 June 2023)) to obtain the latest KEGG Pathway gene annotation as the background. The differential genes were mapped to their respective background sets, and the R software package clusterProfiler (version 3.14.3) [21] was used for enrichment analysis to obtain the results of gene set enrichment. The minimum gene set was set to 5, the maximum gene set was 5000, and *p* < 0.05 was considered significant for enrichment.

### 2.5. Weighted Gene Co-Expression Network Analysis 

Based on the scale-free topology standard, the co-expression network in the expression profile of the dataset GSE144358 was constructed using weighted gene co-expression network analysis (WGCNA) [22]. Firstly, the soft threshold power and adjacency were calculated from the gene expression profile, and outlier genes and samples were removed using the goodSamplesGenes function in the R software WGCNA package. Then, the scale-free co-expression network was constructed by WGCNA. 

The Pearson correlation matrix and the average linkage method were first applied to all pairwise genes. Then, the weighted adjacency matrix was created with the formula A_mn_ = |C_mn_|^β^ (where A_mn_: adjacency between gene m and gene n, C_mn_: Pearson’s correlation, and β: soft-power threshold). After selecting the power of 3, the adjacency relationship was transformed into a topological overlap matrix (TOM), which measures the network connectivity of a gene. TOM is defined as the sum of the adjacency relationships between the gene and all other genes in the network, and the corresponding dissimilarity (1-TOM) is calculated.

To classify genes with similar expression profiles into different gene modules, the minimum module size was set to 50 (genome), and average linkage hierarchical clustering was performed according to the tom-based dissimilarity measure. For module analysis, the sensitivity parameter was set to 3, the module merging threshold was 0.25, and |MM| was 0.8, while |GS| was 0.1.

### 2.6. Identification of Characteristic Genes 

Candidate hub genes were identified by the intersection of DGE and WGCNA key module genes. Then, the hub genes were further screened using LASSO and Random Forest machine learning algorithms.

To integrate survival time, survival status, and gene expression data, we used the R software glmnet package to perform regression analysis using the LASSO–Cox method [23]. We also set up a 10-fold cross-validation to obtain the optimal model. Additionally, we determined the optimal number of variables using the average error rate of candidate hub genes for DEGs in the randomForest package in R software [24]. By calculating the error rate of each tree from 1 to 500 trees, we determined the optimal number of trees based on the lowest error rate and selected the genes with the highest feature importance score as reference genes. 

The two machine learning algorithms were then analyzed by Wayne graph intersection analysis, and the intersection genes were considered characteristic gene combinations of patients with acute SFTS phase. Subsequently, ROC analysis was performed on the obtained characteristic gene combinations using the R software package pROC (version 1.17.0.1). The diagnostic efficiency of these characteristic genes was evaluated based on the area under the curve (AUC) of the ROC characteristic curve (ROC). An AUC greater than 0.75 was selected to indicate that the model had good diagnostic performance.

### 2.7. Gene Set Enrichment Analysis 

To further determine the association between the combination of characteristic genes and signaling pathways, we used GSEA software (version 3.0) obtained through the Gene Set Enrichment Analysis (GSEA) website. The samples were divided into a high expression group (≥50%) and a low expression group (<50%) according to the hub gene expression levels. We downloaded the c2.cp.kegg.v7.4.symbols.gmt subset from the Molecular Signatures Database to evaluate related pathways and molecular mechanisms [25].

Based on gene expression profiles and phenotypic grouping, the minimum gene set was set to 5, the maximum gene set was 5000, and a thousand re-samplings were performed. A *p*-value less than 0.05 was considered statistically significant.

### 2.8. Immune Cell Infiltration 

The immune system of the body encompasses three distinct tiers of defense. The initial tier, known as the first line of defense, comprises the skin and mucosal barrier. Subsequently, the second line of defense is constituted by macrophages and bactericidal substances. Both of those two lines of defense are non-specific immune components. The third line of defense, on the other hand, is a specific immune defense mechanism that encompasses immune organs and immune cells. In the event that the pathogen effectively breaches the first two lines of defense, it will elicit the activation of the third specific immune defense line. At this time, the immune organs and a large number of immune cells will be urgently mobilized to defend against foreign pathogens. CIBERSORT is the tool for deconvolution of the expression matrix of human immune cell subtypes based on the principle of linear support vector regression. By using the chip expression matrix and sequencing expression matrix, the deconvolution analysis of the expression matrix of unknown mixtures and similar cell types is performed. This research method has become the most cited immune cell infiltration estimation analysis tool since it was first published in the Nature method in 2015 [26]. Based on the known reference data set, CIBERSORT compares the gene expression feature sets of 22 immune cell subtypes provided in the database, further performs non-negative matrix decomposition on the expression matrix, and calculates the proportion of different cell types, so as to clarify the composition of immune cells in the human microenvironment, and then clarify which immune cells play an important role in the occurrence and development of the disease. CIBERSORT can be used to explore the difference of immune cell subtypes between the SFTSV acute phase group and the non-infection control group [26]. Therefore, we screened immune cells with significant differences in infiltration between the SFTSV acute phase group and the non-infected control group and used the Spearman method to analyze their correlation with characteristic genes.

### 2.9. Statistical Analysis 

All statistical analyses in this study were conducted using R software (version 4.2.2). Unless otherwise stated, a *p*-value less than 0.05 was considered statistically significant, and all *p*-values were two-tailed.

## 3. Results

### 3.1. Identification of DEGs between SFTSV Acute Phase and Non-Infected Control 

The GEO data set provider showed that there were clear clinical data to distinguish between SFTSV acute patients and healthy controls in the data set introduction. In order to further clarify the difference between SFTSV acute phase patients and healthy people in the data set from the molecular level, this study conducted cluster analysis on the sample set. We first used the R software ‘UMAP’ package for principal component analysis and found that the SFTSV acute phase and non-infected control groups could be significantly distinguished (Figure 2A). As expected, SFTSV acute patients and healthy controls can be effectively distinguished in the results. We then used the “limma” package to analyze the DEGs of these two groups, selecting a 2.0-fold difference and an FDR less than 0.01. A total of 10,309 DEGs were screened, including 8793 up-regulated genes and 1516 down-regulated genes (Figure 2B). We displayed the top 100 DEGs between the SFTSV acute phase and non-infected control groups using a heat map, which included the top 50 up-regulated and top 50 down-regulated DEGs (Figure 2C).

### 3.2. Functional Enrichment Analysis

GO analysis includes three categories: Biological Process (BP), Cell Component (CC), and Molecular Function (MF). These categories are helpful in exploring the biological processes of these DEGs. The results of BP enrichment of DEGs showed that cell cycle ranked first in the significant enrichment ranking, followed by the mitotic cell cycle, mitotic cell cycle process, organelle organization, DNA metabolic process, cell cycle process, mitotic cell cycle phase transition, mitochondrial translation, cell cycle phase transition, and mitochondrial gene expression. Most of these enriched results have a high correlation with the cell cycle, suggesting that the BP of DEGs between the acute SFTSV phase group and the healthy control group may be involved in affecting the cell cycle process of the host, as shown in (Figure 3A). The results of DEGs enrichment in CC were the nucleoplasm, cytosol, mitochondrion, mitochondrial part, organelle envelope, envelope, mitochondrial envelope, chromosome, mitochondrial protein complex, and nuclear part. Except for the cytosol, organelle envelope, and envelope, in these top 10 CC enrichment results, the remaining 7 results are more correlated with nucleus and mitochondria, such as nucleoplasm, mitochondrion, mitochondrial part, mitochondrial envelope, chromosome, mitochondrial protein complex, and nuclear part. Therefore, these CC enrichment results of the DEGs between the acute SFTSV phase group and the health control group indicate that the cell composition of these DEGs may have a high correlation with the nucleus and mitochondria, as shown in (Figure 3B). In addition, the top 10 MF enrichment results are catalytic activity, antigen binding, RNA binding, hydrolase activity acting on acid anhydrides in phosphorus-containing anhydrides, pyrophosphatase activity, nucleoside-triphosphatase activity catalytic activity, acting on RNA, hydrolase activity, and nucleoside phosphate binding (Figure 3C). Among the convergence of these MF enrichments, most of them are related to the activity of various enzymes and the combination of molecules. This may mean that DEGs between the acute SFTSV phase group and the health control group may be participating in the function of activity of various enzymes and the combination of molecules. And the GO overall enrichment results showed that the nucleoplasm, cytosol, mitochondrion, mitochondrial part, organelle envelope, envelope, mitochondrial envelope, chromosome, mitochondrial protein complex, and nuclear part had significantly higher enrichment rankings (Figure 3D). Regarding KEGG analysis, the top 15 enriched pathways were mainly the Cell cycle, Protein processing in endoplasmic reticulum, Proteasome, Parkinson disease, Alzheimer disease, Oxidative phosphorylation, Thermogenesis, Huntington disease, Non-alcoholic fatty liver disease (NAFLD), Spliceosome, Metabolic pathways, mTOR signaling pathway, B cell receptor signaling pathway, Th17 cell differentiation, and T cell receptor signaling pathway (Figure 3E).

### 3.3. Construction of the Weighted Gene Co-Expression Network 

We used WGCNA in R software to analyze the SFTSV acute phase and non-infection control groups and establish a scale-free co-expression network. We constructed a common expression network based on the optimal soft threshold construction, and the soft threshold was determined to be 3. The scale-free index was 0.87, with favorable average connectivity (Figure 4A,B). By calculating the characteristic genes of different modules, we further reduced the module system clustering tree and merged the modules with a distance less than 0.25. After dividing the genes into different modules, genetic clustering trees can be drawn. From the result chart of the clustering tree, we can see that the upper part is the layered clustering tree diagram of the gene, and the lower part is a gene module, that is, the network module. The clustering results indicate that a total of six co-expression modules can be obtained, namely black, purple, brown, turquoise, magenta and ellow modules (Figure 4C,D). Correspondingly, we can see that the closer gene (cluster to the same branch) is divided into the same module. Then we calculated the gene significance (GS) value of the gene module associated with the clinical grouping of the SFTS acute phase and healthy controls and calculated the module feature vector and the gene expression correlation (MM) value according to the truncation criteria (|MM| > 0.8 and |GS| > 0.1), And then, we also calculated the correlation and significant clinical characteristics of the module and drew the correlation heat map. The heat map results showed that the brown module was the highest correlation with the characteristic performance (cor = 0.94, *p* < 0.0001), while the yellow module was the lowest (cor = 0.30, *p* < 0.0001) (Figure 4E). Therefore, we finally selected the brown module with the highest correlation as a key module and its corresponding 2132 genes for further analysis. We further calculated the correlation between each module and the correlation with SFTSV acute phase. The results showed that the brown module was significantly correlated with SFTSV acute phase (cor = 0.94, *p* < 0.0001) (Figure 4F). Therefore, the brown module containing 2132 genes was considered a key module related to SFTSV acute phase. By overlapping the genes in the brown module with 10,309 DEGs (including 8793 up-regulated genes and 1516 down-regulated genes) and 2132 genes contained in the brown module, we obtained four brown module unique genes, 8181 DEGs unique genes and 2128 overlapping genes (Figure 4G).

### 3.4. Feature Genes Were Selected by LASSO–Cox and Random Forest Algorithms

The feature genes of the 2128 intersection key genes were further screened by LASSO–Cox and Random Forest machine algorithms. In the LASSO–Cox analysis, the Lambda value was set to 0.05, and a total of 24 characteristic genes were obtained (Figure 5A,B). The formula of the model is as follows: RiskScore = −0.0697 ∗ ADIPOR1 + 0.0295 ∗ APOBEC3B-0.0722 ∗ BAG1-0.0021 ∗ BCAM + 0.0419 ∗ BUD31 + 0.0310 ∗ CENPO + 0.0716 ∗ E2F2-0.0306 ∗ ENTPD2 + 0.1174 ∗ FAM174B-0.0023 ∗ GPC4 + 0.0005 ∗ H2AC17 + 0.0814 ∗ IAH1 + 0.0440 ∗ IFI27 + 0.0949 ∗ KDELR3-0.0078 ∗ LOC101928037 + 0.0421 ∗ LOC112543491 + 0.01629 ∗ MIR4435-2HG + 0.1121 ∗ NASP + 0.0007 ∗ PLGRBKT ∗ RAIFNTX0.0174. A total of 41 characteristic genes were obtained in random forest analysis (Figure 5C,D). These selected characteristic genes are shown in Table 1. Through the interaction of these two algorithms, four characteristic genes were finally identified, including ADIPOR1, CENPO, E2F2, and H2AC17 (Figure 5E). In order to further confirm the disease specificity of these four characteristic genes obtained from the screening, we also downloaded the dataset of acute measles patients (GSE5808) and the dataset of AIDS patients (GSE140713) with the same detection method for further analysis and comparison. Moreover, these results showed that these four feature genes could effectively distinguish SFTS patients from patients with acute measles and those with AIDS, so we still initially believe that these four feature genes have the specificity of the SFTSV acute phase (Figure 5F,G). Then, we further analyzed these four characteristic genes.

### 3.5. The Diagnostic Efficacy of Characteristic Genes in Predicting SFTSV Acute Phase 

In this study, we further demonstrated the expression results of four differentially expressed genes screened based on machine learning methods. The results presented in Figure 6A suggest that the expression of characteristic genes ADIPOR1, CENPO, E2F2, and H2AC17 is significantly different in different groups (*p* < 0.0001), indicating that the expression of these four characteristic genes was significantly different between the SFTSV acute phase group and the healthy control group. Moreover, the AUC values in the ROC curve results of these four characteristic genes are all 1, indicating that there is at least one threshold in the model, which can perfectly divide positive and negative samples into different groups (Figure 6B). These results also show that these four characteristic genes of the prediction model can effectively distinguish between different groups of samples.

In addition, the diagnostic efficiency of the validation set cohort for predicting the SFTSV acute phase with each characteristic gene was consistent with the results of the test set. As shown in Figure 7A, there were significant differences in the expression levels of ADIPOR1, CENPO, E2F2 and H2AC17 in the samples between the acute phase group and the healthy group (Figure 7A), and the results in the ROC curve corresponding to their respective characteristic genes were also consistent with the results of the test set (Figure 7B).

### 3.6. GSEA Analysis

By analyzing the GSEA results of the differentially expressed genes, we found that the four differentially expressed genes obtained in this study were mainly related to the cell cycle, which was consistent with the results of GO enrichment and KEGG enrichment. The results showed that there was a big difference between the SFTSV acute phase patients and healthy people, which was related to the regulation of host cell cycle. Specifically, GSEA analysis was used to evaluate the signaling pathways related to characteristic genes, and significantly enriched pathways with *p* values < 0.05 were identified (Figure 8A–D). 

The results showed that ADIPOR1 expression was associated with various pathways such as snare interactions in vesicular transport, RNA degradation, toll-like receptor signaling pathway, systemic lupus erythematosus, oocyte meiosis, intestinal immune network for IgA production, cell cycle, p53 signaling pathway, arrhythmogenic right ventricular cardiomyopathy arvc, apoptosis, basal transcription factors, oxidative phosphorylation, hematopoietic cell lineage, etc. While CENPO expression was correlated with pathways such as RNA degradation, protein export, nucleotide excision repair, cell cycle, p53 signaling pathway, intestinal immune network for IgA production, DNA replication, snare interactions in vesicular transport, motor signaling pathway, chronic myeloid leukemia, etc. And the E2F2 was associated with pathways such as cell cycle, peroxisome, RNA degradation, protein export, DNA replication, RNA polymerase, nucleotide excision repair, p53 signaling pathway, cardiac muscle contraction, chronic myeloid leukemia, etc. The H2AC17 was associated with pathways such as p53 signaling pathway, nucleotide excision repair, DNA replication, cell cycle, RNA degradation, proteasome, protein export, intestinal immune network for IgA production, apoptosis, chronic myeloid leukemia, etc.

Based on the results, it is evident that ADIPOR1 enrichment is mainly involved in the immune system function. And the other three genes (CENPO, E2F2, and H2AC17) are involved in regulating the cell cycle. This finding aligns with the previous GO and KEGG enrichment analyses of the overall DEGs, indicating that, to some extent, the cell cycle of the acute SFTSV-infected patient group may be differentially regulated compared to the healthy control group.

In order to further explore and analyze the functional correlations that these four characteristic genes may involve, we carried out all the significantly enriched pathways (*p* < 0.05) of GSEA enrichment of characteristic by UpSetR. This method is to visualize the UpSetR package based on the data set in R language. In these analysis results, the upper columnic diagram represents the number of intersections, and the lower left bar chart represents the size of the collection path; and the dotted diagram of the lower right indicates the specific combination of the overlapping matrix between the set. And the results showed that the common intersection of these four characteristic gene pathways were homologous recombination, progesterone mediated oocyte maturation, ubiquitin mediated proteolysis, RNA degradation, nucleotide excision repair, systemic lupus erythematosus, oocyte meiosis, intestinal immune network for IgA production, cell cycle, folate biosynthesis, p53 signaling pathway, mismatch repair, vibrio cholerae infection, amino sugar and nucleotide sugar metabolism, and oxidative phosphorylation. This also reminds us that the four characteristic genes may participate in important immune and metabolic signaling pathways (Appendix A) which is consistent with the expected results.

### 3.7. Immune Cell Infiltration 

In fact, in previous studies on SFTS, although researchers knew that SFTSV infection would cause host immune cell disorders, there was no study on the overall changes of host immune cells after SFTSV infection. Therefore, in order to determine the effect of the SFTSV acute phase on host immune cells, CIBERSORT was used to analyze the proportion of host immune cells. This study first evaluated the overall immunological characteristics by immune cell infiltration analysis. A total of 22 immune cell marker genes were provided in the official CIBERSORT database, among which NK_activated_cells, Dendritic_resting_cells and Eosinophils three immune cell groups were not significantly enriched in this dataset. Therefore, these three immune cell groups were excluded from the evaluation results. Among 19 common immune cell groups that showed different expression, gamma delta T cells and activated Dendritic cells had a positive correlation coefficient of 0.73, while Monocytes and Macrophages M0 groups showed a significant positive correlation with a correlation coefficient of 0.73. However, gamma delta T cells and CD4 memory resting T cells had a significant negative correlation with a correlation coefficient of −0.83 (Figure 9A).

Interestingly, in this study, we found that SFTSV infection altered host immune infiltration, specifically, the acute SFTSV phase increased the proportion of memory B cells, Plasma cells, CD4 memory activated T cells, regulatory T cells (Tregs), gamma delta T cells, Macrophages M0, Macrophages M1, activated Dendritic cells, resting Mast cells and Plasma cells (Figure 9B). The proportion of naive B cells, CD8 T cells, CD4 memory resting T cells, and resting NK cells was higher in healthy control population tissues (Figure 9B). Moreover, correlation analysis was performed between the four selected characteristic genes and 13 immune cell groups showing significant differential infiltration (*p* < 0.05). The results showed that CENPO, E2F2, and H2AC17 were negatively correlated with the infiltration of naive B cells, CD8 T cells, and CD4 memory resting T cells, but positively correlated with the infiltration of other immune cells. ADIPOR1 was negatively correlated with the infiltration of memory B cells, Plasma cells, CD4 memory activate T cellsd, and gamma delta T cells, but positively correlated with other immune cells (Figure 9C). 

Therefore, the analysis results suggest that the SFTSV acute phase does indeed have some association with immune cell disorders, and the absolute correlation between the four characteristic genes and T cell and B cell infiltration was significantly higher than that of other immune cells. For example, the absolute correlation coefficients between naive B cells, memory B cells, CD8 T cells, CD4 memory resting T cells, CD4 memory activated T cells, and the four characteristic genes ranged from 0.53 to 0.85 (Figure 9C). These results suggest that the changes in the composition of these immune cells may be related to the SFTSV acute phase, which may also be a new idea and direction for future SFTS immunotherapy research.

## 4. Discussion

SFTSV is a tick-borne virus that can cause severe acute fever syndrome. Currently, there are no specific therapies or preventive measures against this virus. Therefore, early diagnosis and treatment are crucial for preventing the spread of SFTSV infection. In addition, the study of biomarkers can also be used to monitor disease progression and treatment response, providing important clues for the development of relevant prevention and treatment strategies. Hence, the importance and urgency of researching biomarkers for the acute SFTSV phase are highlighted.

This study evaluated the differentially expressed genes (DEGs) between individuals with the acute SFTSV phase and healthy individuals, and then performed GO and KEGG enrichment analysis on these DEGs. The results showed significant enrichment of cell cycle and related pathways among the highly ranked terms. Previous research has reported that various viruses are involved in influencing and regulating the host cell cycle upon infection, causing host cell cycle arrest and promoting viral replication. However, no literature on the impact of SFTSV infection on the host cell cycle has been reported. Additionally, this study used WGCNA module analysis and combined DEGs analysis to identify 2128 differentially expressed genes highly associated with acute SFTSV phase. Further analysis using LASSO–Cox and random forest machine learning methods identified four feature genes (ADIPOR1, CENPO, E2F2, and H2AC17) that are significantly associated with SFTSV infection. Subsequently, expression profiles and ROC analysis of these four differential feature genes were examined in both the training dataset and validation dataset, and the results matched the expected GSEA analysis that was conducted on these four feature genes to explore their relevant signaling pathways. The analysis revealed that, apart from ADIPOR1, which primarily contributes to immune system function, the other three genes are involved in regulating the cell cycle. This finding is consistent with the previous GO and KEGG enrichment analysis of overall DEGs, indicating that the cell cycle in patients with the acute SFTSV phase may be regulated differently compared to the healthy control group to some extent. Moreover, in order to investigate the impact of the acute SFTSV phase on host immune cells, this study conducted immune cell infiltration analysis to evaluate the overall immunological characteristics. The CIBERSORT algorithm was utilized to explore and analyze changes in the composition of host immune cells between the acute SFTSV phase group and the healthy control group.

Since the measles virus and HIV infection can both induce the abnormal immune reaction or affect the immune reaction in the host, we also downloaded the PBMC dataset of acute measles patients (GSE5808) and the dataset of AIDS patients (GSE140713) to further confirm the disease specificity of these four characteristic genes obtained from the screening. Moreover, the results showed that these four feature genes could effectively distinguish SFTS patients from patients with acute measles and those with AIDS. Therefore, we still initially believe that these four feature genes have the specificity of the acute SFTSV phase. Then, we further analyzed these four characteristic genes.

Firstly, we elaborated on the potential relationship between the four differentially expressed genes screened in this study and the acute phase of SFTSV. ADIPOR1, also known as Adiponectin Receptor 1, is a gene that encodes a protein primarily expressed in adipose tissue. Its main function is to regulate physiological processes such as energy metabolism and insulin sensitivity by receiving signals from adiponectin. Research has shown that ADIPOR1 not only facilitates energy storage and utilization in the body but also affects lipid metabolism and synthesis, promotes fatty acid oxidation, and has anti-inflammatory effects. Additionally, it is involved in anti-apoptosis, anti-inflammation, anti-fibrosis, anti-lipotoxicity, and promotes increased fibroblast activity [27,28]. Therefore, the down-regulation of ADIPOR1 expression in the infected group may be an important factor contributing to inflammation after viral infection. ADIPOR1 has been identified as one of the main receptors for adiponectin. When adiponectin enters cells, it binds to ADIPOR1, activating a signaling pathway that produces various biological effects. The expression level of ADIPOR1 is regulated by adiponectin, and studies have shown that adiponectin can significantly increase the expression of ADIPOR1 mRNA [29]. Adiponectin can also inhibit the expression of TNF-α and certain adhesion molecules in vitro. Plasma adiponectin levels have been found to decrease in women with mildly elevated C-reactive protein, and low adiponectin levels are negatively correlated with high C-reactive protein [30]. Furthermore, studies have suggested that the C1q receptor on the surface of macrophages is one of the receptors for adiponectin. Adiponectin can significantly inhibit the formation of granulocyte colony-forming unit-granulocyte (CFU-GM), granulocyte colony-forming unit-granulocyte (CFU-G), and monocyte-macrophage colony-forming unit-monocyte (CFU-M) but has no effect on the formation of early erythroblast precursor cells (BFU-E) and mixed erythroblast colonies. Adiponectin can also significantly inhibit the phagocytic activity of mature macrophages, suggesting that the combination of ADIPOR1 and adiponectin may be an important regulator of the inflammatory response. Additionally, ADIPOR1 may be involved in regulating immune responses [31]. 

Abnormally high expression of CENPO can disrupt the regulation of the cell cycle. For example, studies have shown that the CENPO gene is involved in regulating cell proliferation and apoptosis, mainly through its interaction with p53. This interaction may play a role in promoting colorectal cancer through the EMT and PI3K/AKT signaling pathways [32]. Additionally, the function of CENPO in cell cycle regulation may have implications for viral infections. Some viruses rely on cell division for replication, and since CENPO is involved in chromosome division, it may affect viral replication. Therefore, studying CENPO can help uncover the molecular mechanisms of viral infection and provide insights for the development of antiviral drugs. Furthermore, CENPO has been implicated in predicting conflicts between viruses and the human host [33].

E2F2 is a member of the E2F family of DNA-binding transcription factors, which bind to Rb and are released through phosphorylation by cell cycle protein/CDK kinases. E2F2 plays a crucial role in regulating genes involved in cell proliferation, invasion, secretion of inflammatory factors [34] and organismal aging [35]. Additionally, studies have identified E2F2 and H2AFx as key genes in the cell cycle process, while miR-24 plays a role in the terminal differentiation of mammalian hematopoietic stem cells (HSCs) [36]. For instance, research has shown that miR-24 can restrain cell proliferation by targeting E2F2, MYC, and other cell cycle genes. Moreover, E2F2 is crucial for myeloid development [37]. Overall, E2F2 is a multifaceted transcription factor that plays a vital role in various physiological and pathological processes. And understanding the functional mechanisms of E2F2 is also important for investigating, treating, and preventing related diseases.

H2AC17 (H2A clustered histone 17) is a unique histone protein with both N-terminal and C-terminal tails. The C-terminal tail is involved in the structure of highly compacted chromatin. Mutations in H2AC17 can directly affect chromosome structure and cell growth. Histones, including H2A, H2B, H3, and H4, are responsible for the nucleosome structure of chromosomal fibers in eukaryotes. Abnormal changes in H2AC17 can lead to instability and dissociation of nucleosome structure and hinder the formation of higher-order chromatin structures. This can loosen chromosome structure and affect the entry of transcription factors, RNA polymerases, and transcription complexes.

Additionally, the reason why this study designed and analyzed the immune infiltration analysis of acute SFTSV infected and healthy people is that with the advancement of omics technology, more and more researchers have learned that the changes in the host’s immune microenvironment after viral infection have potential research significance for improving immunotherapy. Therefore, further analysis and understanding of the host microenvironment infected by the virus, as well as the analysis of the composition of immune cells in the host immune tissue has gradually become a research hotspot. The essence of analyzing the host immune microenvironment is to analyze the proportion of a large number of aggregated immune cells. The CIBERSORT algorithm adopted in this study can effectively analyze the composition of immune cells in patients with acute SFTSV infection and healthy people. Interestingly, the results of this study found that SFTSV infection was indeed involved in changing the state of host immune infiltration. Specifically, the acute SFTSV phase showed an increase in these immune cells, such as memory B cells, Plasma cells, CD4 memory activated T cells, regulatory T cells (Tregs), gamma delta T cells, Macrophages M0, Macrophages M1, activated Dendritic cells, resting Mast cells and Plasma cells. And the proportion of naive B cells, CD8 T cells, CD4 memory resting T cells, and resting NK cells was higher in healthy control population tissues. Not only that, among the differential genes screened in this study, CENPO, E2F2, and H2AC17 are directly or indirectly involved in the regulation of the host cell cycle, which aligns with the results of enrichment analysis in this study. It is hypothesized that SFTSV may have related regulatory mechanisms after infecting the host, considering the influence and regulation of the host cell cycle following viral infection. During the acute phase of SFTSV infection, innate and humoral immune disorders are significant pathological mechanisms in SFTS. Effective activation and differentiation of myeloid DC cells (mDC) and follicular Th cells (Tfh) play a crucial role in clearing the virus. The inflammatory response in SFTSV-infected patients is characterized by imbalanced secretion of cytokines and chemokines, with Th1 cytokines being correlated with disease severity [38]. In patients with acute phase and those who died, serum cytokines such as IL-1, IL-6, IL-10, colony-stimulating factor G-CSF, IFN-γ inducer protein 10, and monocyte chemoattractants protein-1 were generally increased, while the cytokine contents of PDGF-BB and RANTES were generally decreased, causing a “cytokine storm”. However, these cytokine levels returned to normal after SFTSV patients recovered [38,39,40], indicating that the “cytokine storm” may be one of the causes of severe infection phase and death. Additionally, the levels of platelets, serum enzymes, inflammatory cytokines, and anti-inflammatory cytokines were closely correlated with SFTSV viral load [41], suggesting that an increase in the SFTSV viral load was also correlated with immune cell disorder. Research in 2018 revealed that failure of specific antibody production and B-cell differentiation were important factors contributing to severe illness and death in SFTS infection. The study found that patients with acute severe illness and death had seriously abnormal numbers and function of myeloid dendritic cells (mDC). Furthermore, Tfh cells in these infected individuals not only had fewer numbers, but also lost their ability to stimulate the activation of B cells. Therefore, apoptosis and dysfunction of mDC induced by SFTSV infection are among the main factors causing dysfunction of CD4+ T cells, and dysfunction of Tfh cells in CD4+ T directly leads to dysfunction of B cell activation and failure of antibody type conversion. In this mechanism, Tfh cells act as a bridge between innate and adaptive immune deficiency caused by SFTSV infection [18]. These results indicated that the differential expression of characteristic genes was significantly associated with immune cells, aligning with the actual disease course of the SFTSV acute phase patients. Therefore, the four-feature gene combination identified by the machine learning algorithm in this study is scientifically applicable and can provide valuable references for the clinical diagnosis and prognosis of SFTS in future clinical practices. This study screened the target characteristic genes from a large amount of data in public databases using bioinformatics methods, leveraging the maturity of sequencing technology and the extensive data available. Furthermore, while these four characteristic genes could effectively distinguish SFTS patients from patients with acute measles and those with AIDS, it is also clear that data related to SFTS patients have not been sufficiently compared with data related to patients with measles and HIV, which is a limitation of our current study. Therefore, in future studies, we will continue to conduct more experimental validation and complement other disease types to help better confirm the specificity of these four characteristic genes.

## 5. Conclusions

In conclusion, this study identified four differentially expressed genes (ADIPOR1, CENPO, E2F2, and H2AC17) for the first time as potential markers for diagnosing the SFTSV acute phase for the first time by using machine learning algorithms. The use of the CIBERSORT algorithm also provided insights into immune cell infiltration patterns during the acute SFTSV phase. This research contributes to the understanding of SFTS through omics analysis and provides a new approach for identifying gene markers in disease diagnosis and prognosis. The analysis of immune cell infiltration offers potential implications for clinical diagnosis and immunotherapy of SFTS. In summary, this study provides a new research idea for the clinical diagnosis and immunotherapy of the SFTSV acute phase.

## Figures and Tables

**Figure 1 viruses-15-02126-f001:**
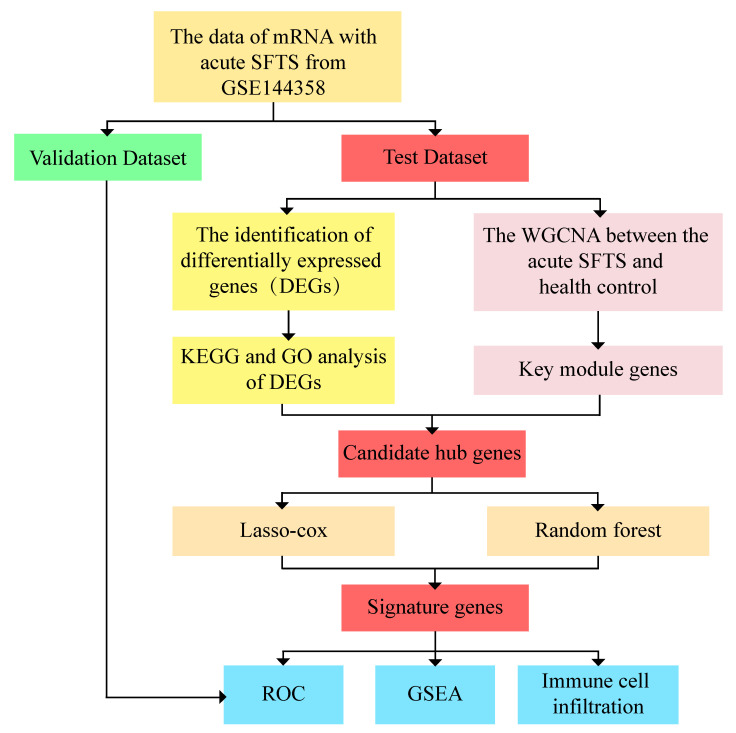
Flowchart of this study.

**Figure 2 viruses-15-02126-f002:**
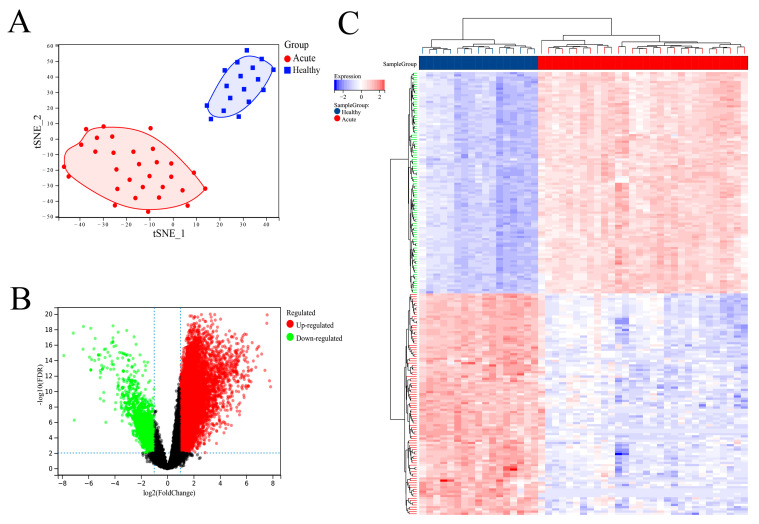
Identification of DEGs. (**A**) Principal component analysis results of the SFTSV acute phase and non-infection control groups. (**B**) Volcano plot of differential genes between the SFTSV acute phase group and non-infection control group. (**C**) Heat maps of differential genes (top 50 up-regulated genes and top 50 down-regulated genes) in the SFTSV acute phase and non-infection control groups.

**Figure 3 viruses-15-02126-f003:**
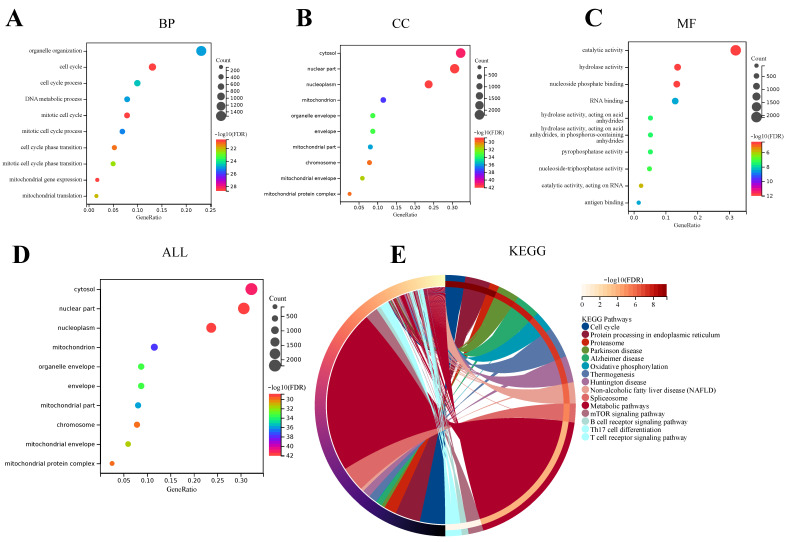
Functional Enrichment Analysis of DEGs. (**A**) GO enrichment results of DEGs in Biological Processes (BP). (**B**) GO enrichment results of DEGs in Cellular Component (CC). (**C**) GO enrichment results of DEGs in Molecular Function (MF). (**D**) GO enrichment results of DEGs overall. (**E**) KEGG enrichment results of DEGs.

**Figure 4 viruses-15-02126-f004:**
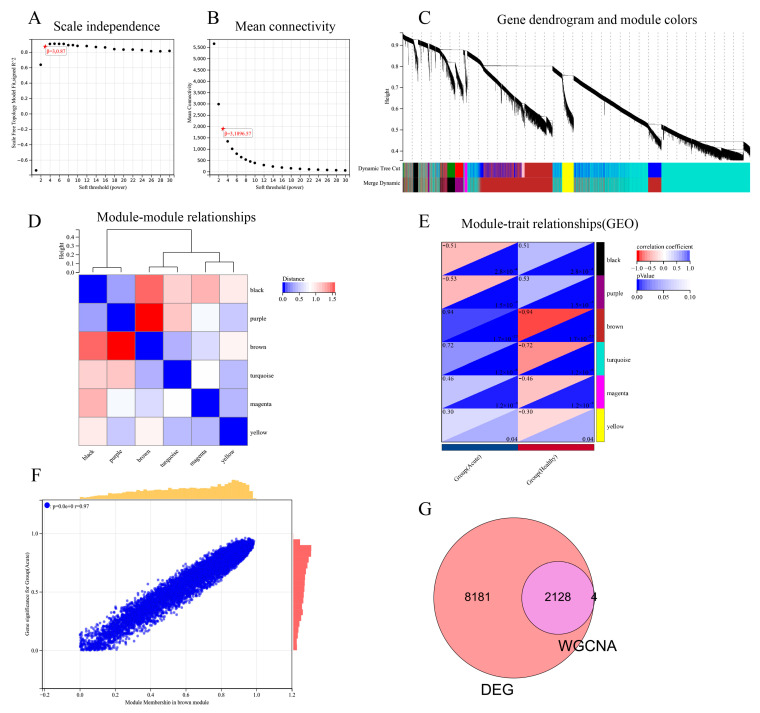
WGCNA module analysis of the GSE144358 dataset and identification of candidate key genes. (**A**) Soft threshold power results of WGCNA. (**B**) Average connectivity results of WGCNA. (**C**) WGCNA clustering tree results. (**D**) Correlation analysis results between WGCNA clustering modules. (**E**) Correlation analysis results between WGCNA clustering module and differential grouping. (**F**) Results of correlation analysis between module membership in brown module and gene significance for Group (Acute). (**G**) Intersection gene results of DEGs and genes in brown module.

**Figure 5 viruses-15-02126-f005:**
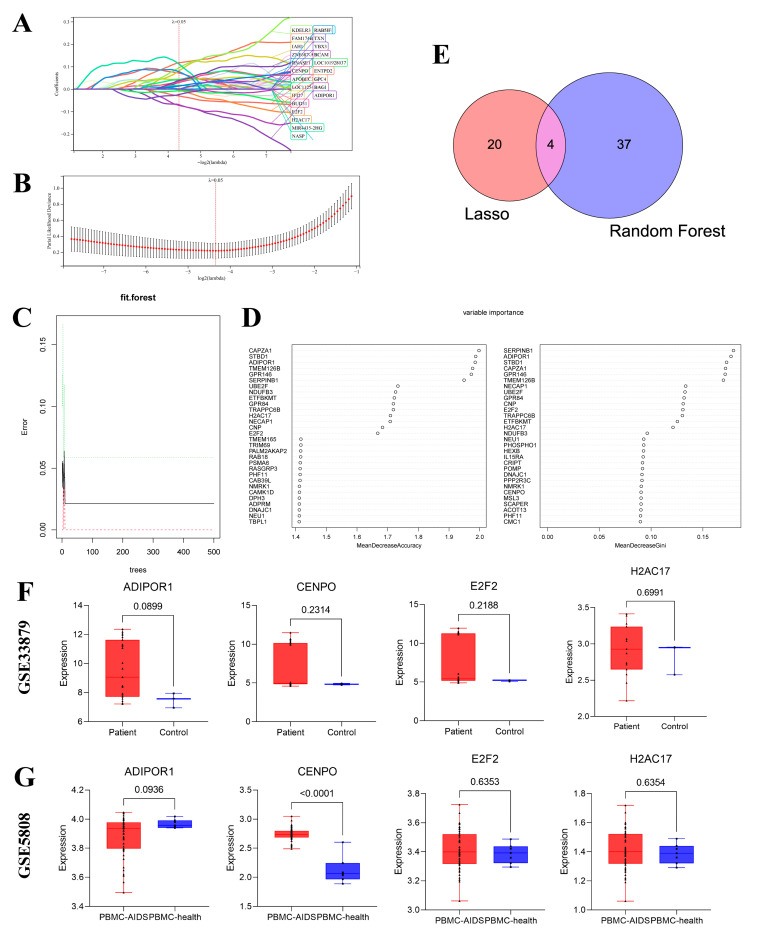
The figure shows the process of obtaining characteristic genes using a machine learning algorithm. (**A**) The LASSO–Cox regression model is used to screen for characteristic genes. (**B**) A fixed lambda value is determined by cross-validation. (**C**) The error rate confidence interval of the random forest model. (**D**) The candidate feature genes are recommended by the random forest algorithm. (**E**) The intersection results of the candidate feature genes obtained by the LASSO–Cox and random forest algorithms. (**F**) The expression of four characteristic genes in patients and health control groups in AIDS. (**G**) The expression of four feature genes in patients and health control groups in Acute measles.

**Figure 6 viruses-15-02126-f006:**
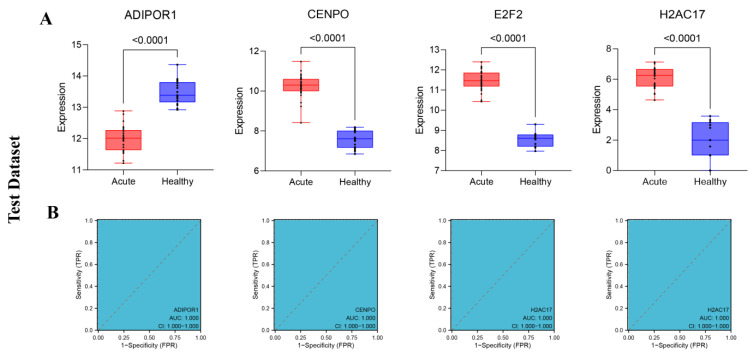
The performance of four characteristic genes obtained from the test data set. (**A**) The expression levels of the four characteristic genes obtained from the test data set between different groups. (**B**) The ROC curve results for the four characteristic genes obtained from the test data set.

**Figure 7 viruses-15-02126-f007:**
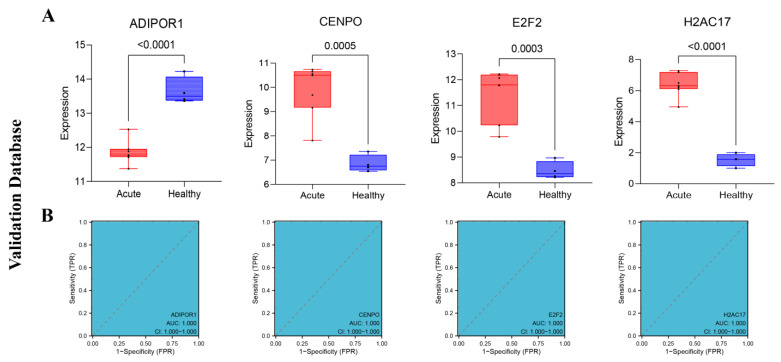
The performance of four characteristic genes obtained from the validation data set. (**A**) The expression levels of the four characteristic genes obtained from the validation data set between different groups. (**B**) The ROC curve results for the four characteristic genes obtained from the validation data set.

**Figure 8 viruses-15-02126-f008:**
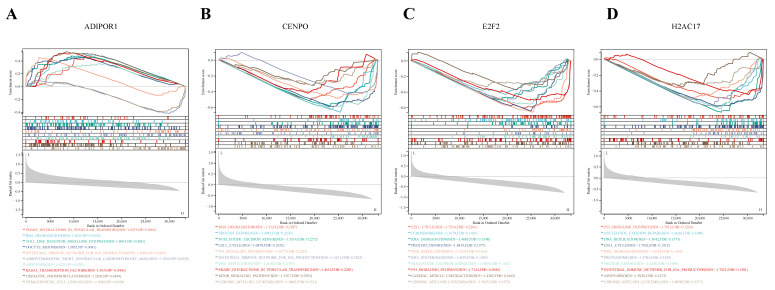
GSEA results of 4 characteristic genes. ADIPOR1 (**A**), CENPO (**B**), E2F2 (**C**) and H2AC17 (**D**) was identified as characteristic genes.

**Figure 9 viruses-15-02126-f009:**
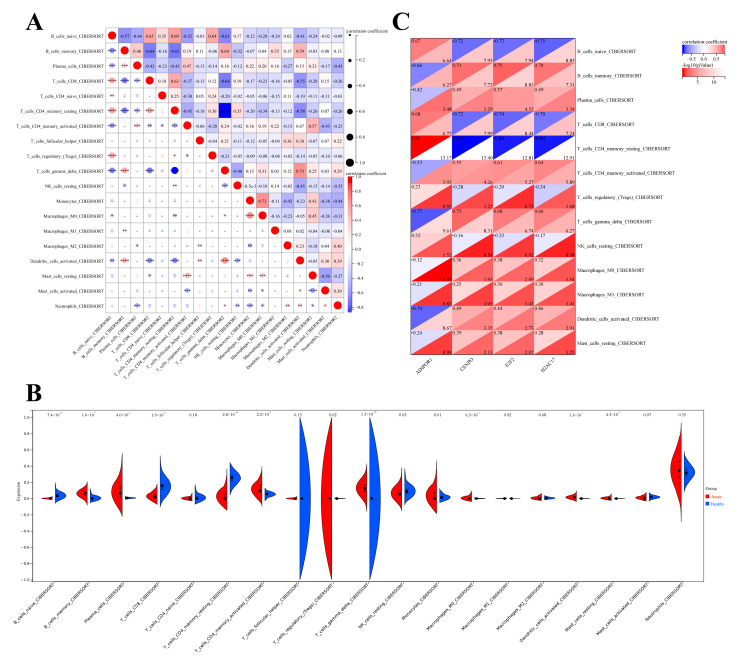
Correlation analysis of immune cell infiltration and 4 characteristic genes. (**A**) The results of different immune cell infiltration. * means *p* < 0.05, ** means *p* < 0.01, *** means *p* < 0.001, **** means *p* < 0.0001. (**B**) Differential expression of cells infiltrated by different immune cells. (**C**) Correlation analysis of immune cell infiltration and 4 characteristic genes.

**Table 1 viruses-15-02126-t001:** Candidate Genes Display Table Screened by LASSO–Cox and Random Forest Analysis.

Lasso–Cox	Random Forest
ADIPOR1, APOBEC3B, BAG1, BCAM, BUD31, CENPO, E2F2, ENTPD2, FAM174B, GPC4, H2AC17, IAH1, IFI27, KDELR3, LOC101928037, LOC112543491, MIR4435-2HG, NASP, PLGRKT, RAB5IF, RNASE1, TXN, YBX3, ZNF687-AS1	TBPL1, NEU1, DNAJC1, ADPRM, DPH3, CAMK1D, NMRK1, CAB39L, PHF11, RASGRP3, PSMA6, RAB18, PALM2AKAP2, TRIM69, TMEM165, E2F2, CNP, NECAP1, H2AC17, TRAPPC6B, GPR84, ETFBKMT, NDUFB3, UBE2F, SERPINB1, GPR146, TMEM126B, ADIPOR1, STBD1, CAPZA1, CMC1, ACOT13, SCAPER, MSL3, CENPO, PPP2R3C, POMP, CRIPT, IL15RA, HEXB, PHOSPHO1

## Data Availability

The original contributions presented in the study are included in the article and Appendix A. Further inquiries can be directed to the corresponding author or the first author.

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
