# Peer review of "Identification and Analysis of a Four-Gene Set for Diagnosing SFTS Virus Infection Based on Machine Learning Methods and Its Association with Immune Cell Infiltration"

_viruses, 2023, doi:10.3390/v15102126_

Round 1

Reviewer 1 Report

In conclusion, this study identified four differentially expressed genes (ADIPOR1, 5CENPO, E2F2, and H2AC17) for the first time as potential markers for diagnosing SFTSV acute infection for the first time by using machine learning algorithms. 

: Authors also check another viral infection such as COVID-19 and show these data. Because most viral infections are similar.

Therefore, after authors compare with gene expression of other viral infections, I accept ADIPOR1, 5CENPO, E2F2, and H2AC17 are potential markers for diagnosing SFTSV acute infection. 

Are ADIPOR1, 5CENPO, E2F2, and H2AC17 connected to cytokines such as IL-10, -6, and TGF-β? 

Because some cytokines are linked to severity of SFTSV infection ((Ref. Fatal outcome of severe fever with thrombocytopenia syndrome (SFTS) and severe and critical COVID-19 is associated with the hyperproduction of IL-10 and IL-6 and the low production of TGF-β. J Med Virol. 2023 Jul;95(7):e28894).

Good

Author Response

Dear reviewers,

Thank you for your letter and the reviewers’ comments concerning our manuscript entitled “Identification and Analysis of A Four-gene Set for Diagnosing Acute SFTSV Infection Based on Machine Learning Methods and Its Association with Immune Cell Infiltration.” (viruses-2557678). We are all honored to get your valuable opinions. Those comments are valuable and very helpful. We have read through comments carefully and have made corrections.Through consulting the paper, we did find that patients with acute infection of SFTSV will cause cytokines disorders, and we all do agree with your point of view. Although we only found PBMC detection data sets in GSE144358 and did not find any data corresponding to the patient's serum cytokine data sets, but we really plan to continue to study the correlation between targets and cytokines. At present, we have started to design the research plan of this new project, and started to contact cooperative in SFTS popular area to prepare for the collection of serum samples, PBMC cell samples and clinical data. Not only that, at your suggestion, we have also supplemented the expression of 4 characteristic genes in host PBMC cells in other diseases, such as acute measles (GSE5808) and AIDS (GSE140713). Based on the instructions provided in your letter, we uploaded the file of the revised manuscript.

I wish you all the best.

All author respects

September 23, 2023

Reviewer 2 Report

Authors obtained four characteristic genes as biomarkers for acute SFTSV infection. It has been thought that SFTSV induces the abnormal immune reaction. Thus, it is plausible that four genes associated with immune function and cell cycle could be markers of acute SFTSV infection. However, the abnormal immune reaction is not specific to SFTSV infection. If authors think these genes could be used as biomarkers for acute SFTSV infection, it is better to compare to other infectious diseases.

Author Response

Dear reviewers,

Thank you for your letter and the reviewers’ comments concerning our manuscript entitled “Identification and Analysis of A Four-gene Set for Diagnosing Acute SFTSV Infection Based on Machine Learning Methods and Its Association with Immune Cell Infiltration.” (viruses-2557678). We are all honored to get your valuable opinions. Those comments are valuable and very helpful. We have verified and modified the article based on your suggestions. Not only that, we have also supplemented the expression of 4 characteristic genes in host PBMC cells in other diseases, such as acute measles (GSE5808) and AIDS (GSE140713) at your suggestion. Based on the instructions provided in your letter, we uploaded the file of the revised manuscript.

I wish you all the best.

All author respects

September 23, 2023

Round 2

Reviewer 2 Report

Authors responded well. However, there are some minor points to be corrected.

Author Response

Dear reviewers,

Thank you very much for your attention and the referee’s evaluation and comments on our manuscript entitled “Identification and Analysis of A Four-gene Set for Diagnosing Acute SFTSV Infection Based on Machine Learning Methods and Its Association with Immune Cell Infiltration.” (viruses-2557678). We have revised the manuscript according to your kind advices and referee’s detailed suggestions.  We sincerely hope this manuscript will be finally acceptable to be published on Viruses. Based on the instructions provided in your letter, we uploaded the file of the revised manuscript.

Thank you so much for all your help and looking forward to hearing from you soon.

Best regards.

All author respects

September 26, 2023
